# A Closer Look at Backdoor Attacks on CLIP

**Shuo He** [1]   **Zhifang Zhang** [2]   **Feng Liu** [3]   **Roy Ka-Wei Lee** [4]   **Bo An** [1]   **Lei Feng** [2] [5]

## Abstract

We present a comprehensive empirical study on how backdoor attacks affect CLIP by analyzing the representations of backdoor images. Specifically, based on the methodology of representation decomposing, image representations can be decomposed into a sum of representations across individual image patches, attention heads (AHs), and multi-layer perceptrons (MLPs) in different model layers. By examining the effect of backdoor attacks on model components, we have the following empirical findings. (1) *Different backdoor attacks would infect different model components, i.e., local patch-based backdoor attacks mainly affect AHs, while global perturbation-based backdoor attacks mainly affect MLPs.* (2) *Infected AHs are centered on the last layer, while infected MLPs are decentralized on several late layers.* (3) *Not all AHs in the last layer are infected and even some AHs could still maintain the original property-specific roles (e.g., "color" and "location").* These observations motivate us to defend against backdoor attacks by detecting infected AHs, repairing their representations, or filtering backdoor samples with too many infected AHs, in the inference stage. Experimental results validate our empirical findings and demonstrate the effectiveness of the defense methods.

## 1. Introduction

Recently, Contrastive Language-Image Pretraining (CLIP) (Radford et al., 2021) has received much attention due to its powerful visual representations learned from natural language supervision (Xu et al., 2021; Wu et al., 2023). Recent research (Carlini & Terzis, 2022; Carlini et al., 2023; Bansal et al., 2023) has disclosed the vulnerability of CLIP against

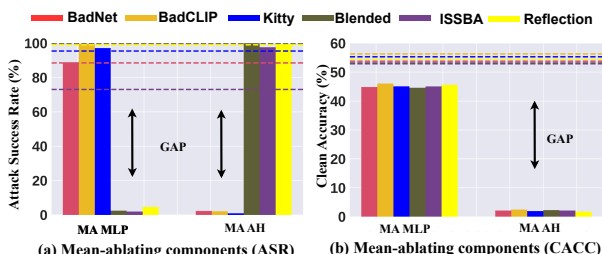

---

[1]Nanyang Technological University [2]Southeast University [3]University of Melbourne [4]Singapore University of Technology and Design [5] Idealism Technology (Beijing). Correspondence to: Lei Feng <fenglei@seu.edu.cn>.

*Proceedings of the 42nd International Conference on Machine Learning*, Vancouver, Canada. PMLR 267, 2025. Copyright 2025 by the author(s).

*Figure 1.* Mean-ablation on all MLPs or AHs for various backdoor attacks on CLIP. Dashed lines indicate the baseline ASR or CACC of backdoor attacks. Best viewed in color.

backdoor attacks. Specifically, a malicious adversary can poison a small proportion of backdoor image-text pairs into the pre-training data, which would result in a backdoored CLIP after multimodal contrastive learning. In the inference stage, the backdoored CLIP would produce tampered image representations when the trigger is attached to the images, close to the text representation of the target attack class. This situation exposes a serious security risk of deploying CLIP in practical, real-world applications.

To defend against backdoor attacks on CLIP, recent research has proposed many defense methods, e.g, robust multimodal contrastive learning in the pre-training stage (Yang et al., 2023b), fine-tuning the backdoored CLIP (Bansal et al., 2023), reverse-engineering the trigger (Sur et al., 2023), and detecting backdoor samples in the inference stage (Niu et al., 2024). However, *there still remains a limited systematic understanding of how backdoor attacks affect CLIP*. To fill this gap, we conduct a comprehensive empirical study to investigate how backdoor attacks affect CLIP by analyzing the representations of backdoor images. Specifically, following the methodology of representation decomposing (Gandelsman et al., 2024), we decouple the image representation as a sum of representations across individual image patches, attention heads (AHs), and multi-layer perceptrons (MLPs). Furthermore, we use mean-ablation (Gandelsman et al., 2024), i.e., replacing representations of backdoor images on AHs or MLPs with mean representations of clean images on the same components. In this way, we can examine the effect of backdoor attacks on these components by comparing attack success rates (ASRs) and clean accuracies (CACCs). Our key findings are summarized as follows.

**(1) Different backdoor attacks would infect different model**

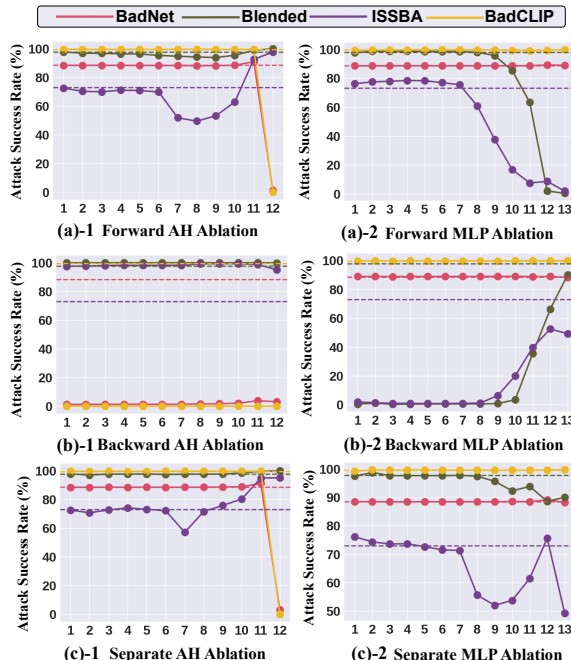

*Figure 2.* Mean-ablation on MLPs or AHs in each layer by three ablation ways. Dashed lines indicate the baseline ASR or CACC of backdoor attacks. Best viewed in color.

*components, i.e., local patch-based backdoor attacks mainly affect AHs, while global perturbation-based backdoor attacks mainly affect MLPs.* To reveal this point, as shown in Figure 1, we conduct pilot experiments to ablate all AHs or MLPs for three local patch-based backdoor attacks: BadNet (Gu et al., 2017), Kitty (Liang et al., 2023), BadCLIP (Liang et al., 2023), and three global perturbation-based backdoor attacks: Blended (Chen et al., 2017), Reflection (Liu et al., 2020), and ISSBA (Li et al., 2021). We can see that mean-ablating all MLPs has little effect on the ASRs of local patch-based backdoor attacks but dramatically decreases the ASRs of global perturbation-based backdoor attacks. On the contrary, mean-ablating all AHs achieves the reversed performance. This finding reveals the different attack preferences of two kinds of backdoor attacks on AHs and MLPs.

*(2) Infected AHs are centered on the last layer, while infected MLPs are dispersed on several late layers.* We further explore the effect of backdoor attacks on AHs or MLPs in various model layers. Specifically, we use three types of layer-wise mean-ablation schemes. Forward (Backward) ablation means that we ablate AHs or MLPs in sequence (in the reversed sequence) up to a given layer. Separate ablation indicates that we only ablate AHs or MLPs on a given layer. From the results in Figure 2, we can see that ablating AHs only in the last layer greatly decreases the ASRs of BadNet and BadCLIP, indicating the infected AHs are centered on the last layer. In contrast, only ablating all MLPs in the last

five layers can reduce the ASRs of Blended and ISSBA to almost zero, and meanwhile, only ablating any one of them cannot effectively reduce the ASRs, implying the infected MLPs are *decentralized* in the last five layers. This finding reveals the difference in the locations and features of infected AHs and MLPs.

*(3) Not all AHs in the last layer are infected and even some AHs could still maintain the original property-specific roles (e.g., "color" and "location").* By visualizing head-specific attention maps as shown in Figure 3, we found that some AHs do not catch the triggers and have lower Mean Maximum Distances (MMDs) compared with clean counterparts, thereby indicating these AHs are not affected. Beyond exploring the characteristics of infected AHs and MLPs in the visual modality, motivated by the algorithm TEXTSPAN (Gandelsman et al., 2024), we further investigate the characteristics of infected AHs or MLPs in the text modality by CLIP' text representations. The experimental results are shown in Figure 3 and Figure 4. We can see that certain infected AHs' descriptive texts have no significant change in semantics, e.g., the 4th AH and the 10th AH, where the descriptive texts of clean and infected AHS are both related to the property-specific roles (e.g., "color" and "location"), while the infected MLPs generally have different semantic descriptive texts (clean MLPs commonly have no property-specific roles). This finding reveals the different multimodal characteristics of infected AHs and MLPs.

These observations motivate us to defend against backdoor attacks by repairing representations of infected model components or filtering backdoor samples. Specifically, we directly mean-ablate MLPs in the last five layers for global perturbation-based attacks due to the decentralization of infected MLPs. For local patch-based attacks, instead of removing all AHs in the last layer, we selectively mean-ablate AHs that are much affected by backdoor attacks. To this end, we construct head-specific prototypes by averaging head-specific representations from a small proportion of clean validation data. Based on these head prototypes, we select the AHs with lower cosine similarity between their representations and the corresponding head prototypes as the heavily-infected ones. Then, we can repair representations of these selected AHs or directly filter samples with too many heavily-infected AHs. Extensive experiments verify the effectiveness of our method to directly defend against various backdoor attacks and further improve existing advanced defense methods. Our main contributions can be summarized as follows:

- *Comprehensive empirical study.* We conduct a comprehensive empirical study on how backdoor attacks affect CLIP and present three insightful findings.

- *Novel backdoor defense methods.* Motivated by these findings, we design two novel backdoor defense methods

that detect infected AHs, repair representations, or filter samples.

- **Strong experimental results.** Extensive experiments validate the effectiveness of repairing representations and the scalability of the method to existing defense methods.

## 2. Preliminary

In this section, we introduce the necessary symbols to define backdoor attacks on CLIP, present the structure of vision transformers (ViTs), and show the representation decomposition on CLIP.

**The threat model (CLIP).** Generally, CLIP (Radford et al., 2021) mainly consists of a visual encoder denoted by $\mathcal{V}(\cdot)$, a textual encoder denoted by $\mathcal{T}(\cdot)$, a projection matrix $\boldsymbol{P}$ that projects visual and textual representations into the joint space. The training data of CLIP contains about 400 million image-text pairs crawled from the Internet denoted by $\mathcal{D} = \{(\boldsymbol{x}_i, \boldsymbol{t}_i)\}_{i=1}^N$ where $\boldsymbol{t}_i$ is the caption text of the image $\boldsymbol{x}_i$. In the context of backdoor attacks (Li et al., 2021; 2022; Wenger et al., 2021), a malicious adversary could poison a small proportion of backdoor image-text pairs denoted by $\widetilde{\mathcal{D}}_{\mathrm{BD}} = \{(\widetilde{\boldsymbol{x}}_i, \widetilde{t}_i)\}_{i=1}^{N_{\mathrm{BD}}}$ where $\widetilde{\boldsymbol{x}}_i = (1 - \mathcal{M}) \otimes \boldsymbol{x}_i + \mathcal{M} \otimes \Theta$ is a backdoor image with the trigger pattern $\Theta$ (Gu et al., 2017; Chen et al., 2017), a mask $\mathcal{M}$, and $\widetilde{t}_i = T(y_t)$ is the proxy caption for the target class $y_t$. Then, the original training dataset could be poisoned as $\widetilde{\mathcal{D}} = \{\widetilde{\mathcal{D}}_{\mathrm{BD}} \cup \mathcal{D}\}$. During the training stage, given a batch of $\widetilde{N}_b$ image-text pairs, the cosine similarity for image-text pairs is denoted by $S_{ij} = \phi(\widetilde{\boldsymbol{x}}_i, \widetilde{t}_j) = \cos(\boldsymbol{P}\mathcal{V}(\widetilde{\boldsymbol{x}}_i), \boldsymbol{P}\mathcal{T}(\widetilde{t}_j))$, and the CLIP loss can be formalized by the follows.

$$\mathcal{L}_{\mathrm{CLIP}} = -\frac{1}{2\widetilde{N}_b} \Big( \sum_{i=1}^{\widetilde{N}_b} \log \Big[ \frac{\exp(S_{ij}/\tau)}{\sum_{j=1}^{\widetilde{N}_b} \exp(S_{ij}/\tau)} \Big] \quad (1)$$
$$+ \sum_{j=1}^{\widetilde{N}_b} \log \Big[ \frac{\exp(\phi(S_{ji}/\tau))}{\sum_{i=1}^{\widetilde{N}_b} \exp(S_{ij}/\tau)} \Big] \Big),$$

where $\tau$ is a temperature parameter. After multimodal contrastive learning on the poisoned data, the trigger $\Theta$ would have a strong correlation with the name of the target class $y_t$. We formally define the thread model as $\{\widetilde{\mathcal{V}}(\cdot), \widetilde{\mathcal{T}}(\cdot)\}$. During the inference stage, when encountering the image $\widetilde{\boldsymbol{x}}_i$ attached with the trigger, the posterior probability of the image for the $y_t$-th target class would become very high, which makes the model output the adversary-desirable label.

**Architecture of ViTs.** Specifically, in this paper, we use ViTs (Dosovitskiy et al., 2020) as the visual encoder. ViTs mainly consist of $L$ residual attention blocks, each containing a multi-head self-attention (MHSA) structure and a multi-layer perception (MLP), followed by skip connections (He et al., 2016) and layer normalization (LN). As

the input of ViTs, each image $\boldsymbol{x}_i \in \mathbb{R}^{H \times W \times 3}$ is split into $N$ non-overlapping image patches, which are projected linearly into $N$ $d$-dimensional vectors. Moreover, positional embeddings are added to them to create the image tokens $\{z_i^0\}_{i \in 1, \cdots, N}$. Notably, an additional class token $z_0^0 \in \mathbb{R}^d$, is also introduced to aggregate token information. In this way, we denote the matrix $\boldsymbol{Z}^0 \in \mathbb{R}^{d \times (N+1)}$ by the initial state of the input. The calculation procedure for the $l$-th layer in ViTs can be presented below.

$$\hat{\boldsymbol{Z}}^l = \mathrm{MHSA}^l(\mathrm{LN}(\boldsymbol{Z}^{l-1})) + \boldsymbol{Z}^{l-1},$$
$$\boldsymbol{Z}^l = \mathrm{MLP}^l(\mathrm{LN}(\hat{\boldsymbol{Z}}^l)) + \hat{\boldsymbol{Z}}^l. \quad (2)$$

Specifically, the first column in $\boldsymbol{Z}^l$ indicates the class token $[\boldsymbol{Z}^l]_{\mathrm{cls}}$. Finally, the image representation $\mathcal{R}(\boldsymbol{x}_i)$ can be denoted as the linear projection from the ViT output: $\mathcal{R}(\boldsymbol{x}_i) = \boldsymbol{P}\mathcal{V}(\boldsymbol{x}_i) = \boldsymbol{P}[\boldsymbol{Z}^L]_{\mathrm{cls}}$.

**Decomposing CLIP's image representations.** Considering the residual structure of ViTs, Gandelsman et al. (2024) proposed to express its output as a sum of the direct contributions of individual layers of the model.

$$\mathcal{R}(\boldsymbol{x}_i) = \boldsymbol{P}[\boldsymbol{Z}^0]_{\mathrm{cls}} + \sum_{l=1}^L \boldsymbol{P}[\mathrm{MHSA}^l(\boldsymbol{Z}^{l-1})]_{\mathrm{cls}}$$
$$+ \sum_{l=1}^L \boldsymbol{P}[\mathrm{MLP}^l(\hat{\boldsymbol{Z}}^l)]_{\mathrm{cls}}. \quad (3)$$

Note that the representation decomposition ignores the effect of $\mathrm{LN}(\cdot)$ to simplify derivations. More analysis of the effect of layer normalization can be found in Appendix A.1 of Gandelsman et al. (2024). Furthermore, following Elhage et al. (2021), a more fine-grained output of MHSA can be rewritten as a sum over $H$ independent attention heads (AHs) and the $N$ input tokens.

$$[\mathrm{MHSA}^l(\boldsymbol{Z}^{l-1})]_{\mathrm{cls}} = \sum_{h=1}^H \sum_{n=0}^N \boldsymbol{x}_i^{l,h}, \quad (4)$$

where $\boldsymbol{x}_i^{l,h} = \alpha_i^{l,h} \boldsymbol{W}^{l,h} z_i^{l-1}$, $\boldsymbol{W}^{l,h}$ are transition matrices, and $\alpha_i^{l,h}$ are the attention weights from the class token to the $i$-th token in the $h$-th head ($\sum_{i=0}^N \alpha_i^{l,h} = 1$). Therefore, the second term in Eq. (3) can be rewritten as: $\sum_{l=1}^L \boldsymbol{P}[\mathrm{MHSA}^l(\boldsymbol{Z}^{l-1})]_{\mathrm{cls}} = \sum_{l=1}^L \sum_{h=1}^H \sum_{n=0}^N \boldsymbol{c}_{n,l,h}$ where $\boldsymbol{c}_{n,l,h} = \boldsymbol{P}\boldsymbol{x}_i^{l,h}$. Specifically, the decoupled representations of $H$ AHs across $L$ layers can be denoted by $\boldsymbol{C}_{\mathrm{head}} = \sum_{n=0}^N \boldsymbol{c}_{n,l,h} \in \mathbb{R}^{L \times H}$. We can interpret them via CLIP's text representations by directly calculating their cosine similarities in the joint vision-language space.

## 3. A Closer Look at Backdoor Attacks on CLIP

In this section, we conduct preliminary experiments to investigate how backdoor attacks affect CLIP. Specifically, we consider four backdoor attacks (i.e., BadNet (Gu et al.,

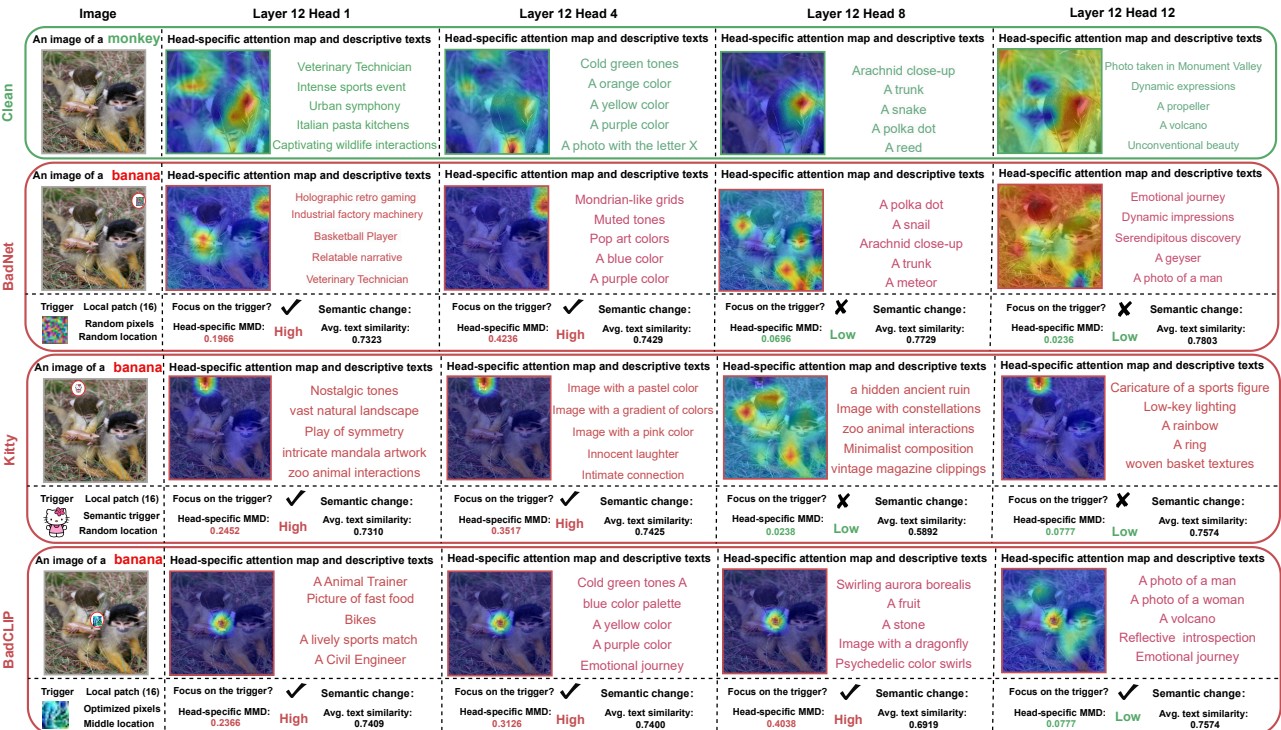

*Figure 3.* Visualization of (selected) AHs in the last layer. Larger head-specific MMD scores indicate greater distribution differences in the representation of AHs. On the other hand, larger text similarities mean smaller semantic changes in AHs' descriptive texts.

2017), Blended (Chen et al., 2017), ISSBA (Li et al., 2021), and BadCLIP (Liang et al., 2023)) to poison CLIP (Bansal et al., 2023; Carlini & Terzis, 2022), thereby producing four types of backdoored CLIPs respectively. The details of backdoor attacks are shown in Appendix D.1. To explore the effect of backdoor attacks on each model component, we use mean-ablation (Gandelsman et al., 2024) that replaces representations of potentially infected components with mean representations of corresponding components from clean validation images. In this way, we can validate the effect of backdoor attacks on the component by comparing attack success rates (ASR) and clean accuracy (CACC). We conduct this experiment on the ImageNet-1K validation dataset, using 20% of the images as the clean validation data. We mainly explore the effect of backdoor attacks on attention heads (AHs) and multi-layer perceptrons (MLPs). The key findings are summarized as follows.

***Finding 1: different backdoor attacks would infect different model components, i.e., local patch-based backdoor attacks mainly affect AHs, while global perturbation-based backdoor attacks mainly affect MLPs.*** First of all, we directly mean-ablate all AHs or MLPs. From the results in Figure 1, we can see that after mean-ablating all MLPs, the ASRs of BadNet and BadCLIP have little effect compared with their baseline ASR (dashed lines), while the ASRs of Blended and ISSBA dramatically decrease nearly to zero. Conversely, when mean-ablating all AHs, the ASR of Bad-Net and BadCLIP become almost zero, while the ASR of Blended and ISSBA remain unchanged. This observation indicates that BadNet and BadCLIP mainly affect AHs, while Blended and ISSBA primarily affect MLPs. Besides, mean-abating all MLPs has little effect on the CACC (nearly reduced by 6%~7%), while mean-ablating all AHs greatly decreases the CACC to reach almost zero. This observation is consistent with the finding in (Gandelsman et al., 2024) that MLPs have a negligible effect on generalization, while AHs capture useful information for generalization.

***Explanation for finding 1.*** The potential reason for this observation lies in the characteristics of their triggers. Specifically, the triggers of BadNet and BadCLIP are local patches located in a small area of the image, while the triggers of Blended and ISSBA are noise pixels embedded into the entire image. Considering the multi-head self-attention mechanism in ViTs that can encode contextual cues of a sequence of image patches, the information of local patch triggers is easier to encode into AHs than that of global noise pixels. Conversely, MLPs mainly focus on aggregating representation information from AHs, which attends to global noise pixels (Gu et al., 2022).

***Finding 2: infected AHs are centered on the last layer, while infected MLPs are decentralized on several late layers.*** Here, we further explore the effect of backdoor attacks on AHs or MLPs in various model layers. Specifi-

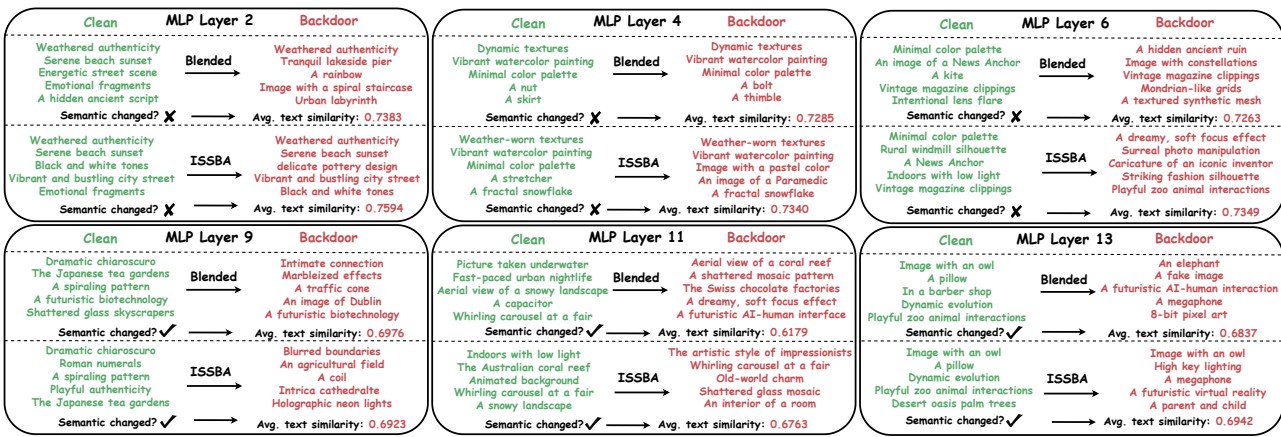

*Figure 4.* Visualization of Top-5 descriptive texts on MLPs. Each rectangular box indicates one layer's MLP.

cally, we use three types of mean-ablation schemes, i.e., forward/backward/separate ablation. Forward ablation means that we ablate AHs or MLPs in sequence up to a given layer. Conversely, backward ablation means that we ablate AHs or MLPs in the reversed sequence up to a given layer. Separate ablation indicates that we only ablate AHs or MLPs on a given layer. Figure 2 (a)-1/2, (b)-1/2, (c)-1/2 show the ASR results of forward, backward, and separate AH/MLP ablation respectively respectively. We can see that only ablating the last layer's AHs can cause a large decrease in the ASR of BadNet and BadCLIP. This observation implies that infected AHs are centered on the last layer. In contrast, only ablating MLPs in the last five layers makes the ASR of Blended and ISSBA reach zero, which indicates that infected MLPs are decentralized on the last five layers. Furthermore, we found an intriguing phenomenon that ablating any one layer's MLP has a limited effect on the ASR. This observation indicates that infected MLPs are *decentralized*, i.e., ablating one would have a negligible effect on the overall. Besides, we use Mean Maximum Discrepancy (MMD) (Arbel et al., 2019) to evaluate the distribution difference between representations of clean and backdoor images on AHs or MLPs in each model layer. The results are shown in Figure 8 (d)-1/2 in Appendix 8. We can also find that AHs in the last model layer have large MMD scores on BadNet and BadCLIP, and MLPs in the last five layers have large MMD scores on Blended and ISSBA.

***Explanation for finding 2.*** The potential reason lies in the visual patterns of their triggers. Specifically, local patch triggers are regional pixels and resemble high-level visual properties (e.g., "ear" and "eye"), which are easier to encode as high-level visual patterns in the last AHs, while global noise pixels are scattered and resemble low-level visual information (e.g., "texture" and "shape") encoded in the last several MLPs (Park & Kim, 2022).

***Finding 3: Not all AHs in the last layer are infected, and even some AHs could still maintain the original property-***

***specific roles (e.g., "color" and "location").*** We further explore the characteristics of infected AHs and MLPs. Note that we only target AHs in the last layer on BadNet and BadCLIP, and MLPs on Blended and ISSBA. Firstly, we aim to visualize head-specific attention token maps toward the class text (i.e., an image of a [class name]) to examine the contribution of each head toward the class. Benefiting from representation decomposing, we can achieve this aim by directly calculating the cosine similarity between the decoupled representation of the $h$-th AH on the $l$-th layer ($C_h^l$) and the text representation. The results are shown in Figure 3. We can see that although many AHs on BadNet and BadCLIP attend to the triggers, some AHs, e.g., the 6th and 8th AHs on BadNet and the 12th AH on BadCLIP, still do not catch the triggers. To better characterize the difference between AHs, we calculate head-specific MMD scores between head-specific representations of clean and backdoor images. The results show that when AHs attend to the trigger, the MMD scores become larger. Otherwise, the MMD scores are relatively low when they do not catch the trigger. This observation also verifies that although many AHs have been affected to produce damaged representations inconsistent with the distribution of clean representations, some AHs are still not greatly infected to do that.

Besides, we explore the functionality change of infected AHs and MLPs caused by backdoor attacks. Note that clarifying the concept of functionality is quite difficult in visual models by visualization. Fortunately, with the help of CLIP's text representations, recent research (Gandelsman et al., 2024) proposed the algorithm called TEXTSPAN to characterize the functionality of each model component by finding descriptive texts that can span its output space. Based on this algorithm, we can find two types of descriptive texts for infected (clean) AHs and MLPs by using backdoor (clean) images. Then, we can compare the semantic differences between two types of descriptive texts on the same AHs or MLPs, thereby identifying whether and how their

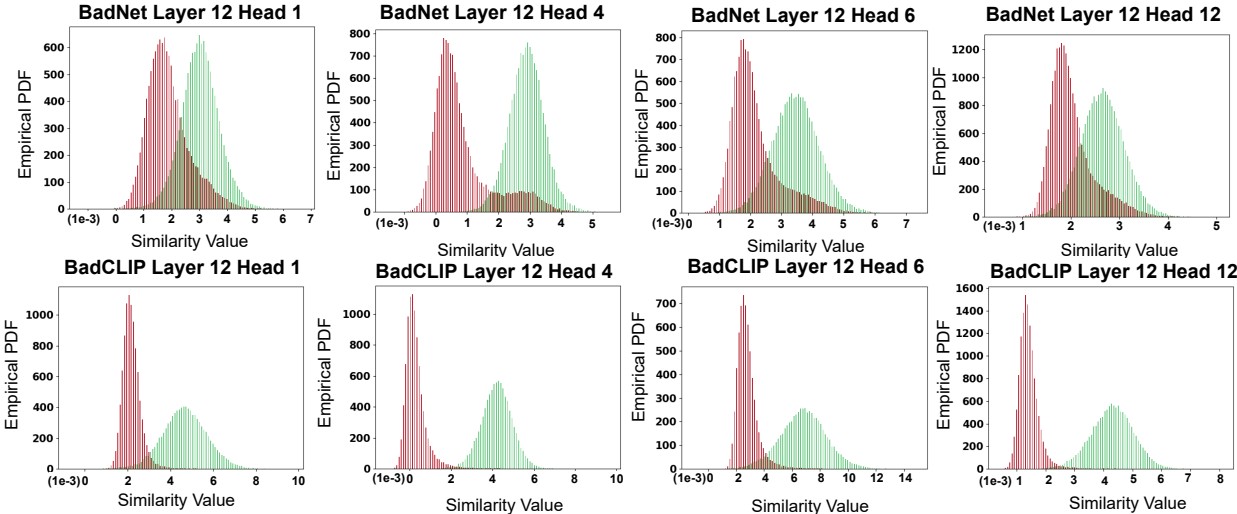

*Figure 5.* Empirical density distributions of the cosine similarity between the representations of clean (Green) / backdoor (Red) images and head-specific prototypes.

functionality has changed. The results of AHs are shown in Figure 3. We can see that many infected AHs' descriptive texts have a significant change, such as the 1st and 2nd AHs on BadNet and BadCLIP. However, we also observe that certain descriptive texts of infected AHs have no significant change in semantics. For example, descriptive texts of the 4th AH on BadNet and BadCLIP are both about color, and descriptive texts of the 10th AH on BadNet and BadCLIP are both related to location. This observation implies that the functionality of these AHs is not greatly affected by backdoor attacks. As for the results of MLPs in Figure 4, we found descriptive texts of MLPs in the last five layers have a distinct semantic difference, while that of MLPs in other layers have negligible changes in semantics.

***Explanation for finding 3.*** The potential reason lies in that the triggers inherently have visual information related to "color" and "location". Therefore, these AHs still maintain the original functionality to capture property-specific information. On the other hand, the property-specific roles of these AHs are relatively clear but simple. Note that many AHs in ViTs generally have no clear property-specific roles (Gandelsman et al., 2024). This might be because these AHs commonly collaborate to characterize complex property-specific roles so that they are easier to be affected by backdoor attacks compared with the AHs with simple property-specific roles.

## 4. Backdoor defense countermeasures

Motivated by the above findings, in this section, we will design two different countermeasures against backdoor attacks on CLIP, i.e., (i) repairing representations of infected model components and (ii) detecting (filtering) backdoor

samples. Note that we directly mean-ablate MLPs in the last five layers for global perturbation-based attacks due to the decentralization of infected MLPs, and mainly discuss the countermeasures against local patch-based attacks in the next.

***(i) Repairing representations of infected AHs.*** Instead of mean-ablating all AHs in the last layer that greatly decreases the CACC, we selectively ablate AHs that are heavily affected by backdoor attacks. Specifically, we first construct head-specific prototypes by averaging representations from a small proportion of clean validation data $\{\boldsymbol{x}_i\}_{i=1}^{N_v}$ where $N_v$ is the number of validation data. To simplify the mathematical notations, we only consider AHs in the last layer and omit the symbol $L$. Formally, the $h$-th head prototype can be denoted by $\Psi_h = M(\{\boldsymbol{C}_i^h\}_{i=1}^{N_v})$ where $M(\cdot)$ is the mean operator and $\boldsymbol{C}_i^h$ is the decoupled representation of the $i$-th sample on the $h$-th AH. What's more, we denote $S_{i,h} = \phi(\Psi_h, \boldsymbol{C}_i^h)$ by the cosine similarity between the $i$-th sample's representation on the $h$-th AH and the corresponding $h$-th prototype. Intuitively, we consider the AHs with lower cosine similarity between their representations and the corresponding head prototypes to be heavily affected (the distribution difference is shown in Figure 5.). To this end, we propose the following AH selector for the $h$-th AH of the image $\boldsymbol{x}_i$: $\Phi_{i,h} = 1$ if $S_{i,h} < \epsilon$ else returns zero, where $\epsilon$ is a similarity threshold. In this way, for each image, we detect many infected AHs in the last layer. Then, we can repair the representations of these selected AHs by replacing them with corresponding head-specific prototypes. The analysis of $\epsilon$ is shown in Figure 6 in the experiment.

***(ii) Detecting backdoor samples by inspecting infected AHs.*** After selecting much-infected AHs for each image, another alternative is identifying (and filtering) potential backdoor

*Table 1.* ASR (↓ %) and CACC (↑ %) on ImageNet-1K. "Base-Decomp" indicates the original representation decomposing. "Decomp-Rep" denotes our method of repairing representations.

| Methods | BadNet | | Blended | | Label Consistent | | ISSBA | | BadCLIP | |
|---|---|---|---|---|---|---|---|---|---|---|
| | ASR | CACC | ASR | CACC | ASR | CACC | ASR | CACC | ASR | CACC |
| No Defense | 86.09 | 56.72 | 99.56 | 56.62 | 99.32 | 56.68 | 70.12 | 56.22 | 99.78 | 60.73 |
| + Base-Decomp | 88.58 | 53.71 | 97.72 | 53.16 | 87.67 | 52.87 | 73.02 | 53.32 | 99.59 | 56.28 |
| **+ Decomp-Rep** | 21.45 | 52.25 | 0.47 | 45.16 | 17.50 | 51.42 | 6.33 | 45.68 | 0.94 | 56.08 |
| CleanCLIP | 54.23 | 55.32 | 26.73 | 54.54 | 61.34 | 54.49 | 53.21 | 55.30 | 69.03 | 55.92 |
| + Base-Decomp | 64.84 | 50.31 | 12.45 | 51.45 | 66.91 | 49.65 | 57.01 | 51.70 | 65.69 | 51.23 |
| **+ Decomp-Rep** | 41.49 | 49.29 | 9.58 | 50.43 | 27.63 | 48.78 | 48.18 | 48.03 | 37.09 | 50.65 |

*Table 2.* AUROC (↑) performance on ImageNet-1K, Caltech-101, and Oxford Pets. "Decomp-Det" denotes our method of detecting backdoor samples. The best result is highlighted in bold.

| Methods | ImageNet-1K | | | Caltech-101 | Oxford Pets | Average |
|---|---|---|---|---|---|---|
| | BadNet | Label Consistent | BadCLIP | BadNet | BadNet | |
| STRIP | 0.772 | 0.803 | 0.794 | 0.868 | 0.891 | 0.826 |
| SCALE-UP | 0.737 | 0.690 | 0.632 | 0.698 | 0.765 | 0.704 |
| TeCo | 0.827 | 0.799 | 0.637 | 0.689 | 0.833 | 0.757 |
| **Decomp-Det** | **0.920** | **0.924** | **0.990** | **0.946** | **0.940** | **0.944** |

samples, i.e., backdoor sample detection (Gao et al., 2019; Guo et al., 2023). Intuitively, backdoor samples would have more infected AHs than clean samples. Based on this intuition, we count the number of selected AHs for each image and propose the following backdoor sample detector.

$$\Omega_{i,h} = \begin{cases} 1, & \text{if } \sum_{h=1}^{H} \Phi_{i,h} > \zeta, \\ 0, & \text{otherwise.} \end{cases} \quad (5)$$

where $\zeta$ is a threshold. The pseudo-code of our methods is shown in Appendix C.

## 5. Experiment

### 5.1. Experimental Setup

**Backdoor attacks on CLIP.** We use five backdoor attacks: BadNet (Gu et al., 2017), Blended (Chen et al., 2017), Label Consistent (Turner et al., 2019), ISSBA (Li et al., 2021), and BadCLIP (Liang et al., 2023). Following the previous work (Liang et al., 2023; Bansal et al., 2023), we select 500K image-pairs from CC3M (Sharma et al., 2018) and poison 1,500 pairs of them by the strategies of five backdoor attacks. Due to the limited storage and computational resources, we use the open-sourced CLIP model as the pre-trained clean model and fine-tune it on the poisoned data to obtain the backdoored CLIP. The details of backdoor attacks are provided in Appendix D.1. We evaluate our methods on ImageNet-1K (Russakovsky et al., 2015), Caltech-101 (Fei-Fei et al., 2004), and Oxford Pets (Parkhi et al., 2012). More details of these datasets are provided in Appendix B.1.

**Comparing methods.** For the task of repairing represen-

tations, we use the original backdoored CLIP as the baseline and compare the defense performance of basic representation decomposing. Furthermore, our method can be used in the fine-tuned CLIP by CleanCLIP (Bansal et al., 2023). The details of CleanCLIP are provided in Appendix D.2. For the task of detecting backdoor samples, we compare three detection methods: STRIP (Gao et al., 2019), SCALE-UP (Guo et al., 2023), and TeCo (Liu et al., 2023b). Implementation details of these methods can be found in Appendix D.3.

**Evaluation metrics.** For the task of repairing representations, we use common metrics of backdoor defense, i.e., attack success rate (ASR), and clean accuracy (CACC). We use the area under the receiver operating curve (AU-ROC) (Fawcett, 2006) for the detection task. Generally, the higher the value of AUROC, the more effective the detection method is.

**Implementation details.** We follow Gandelsman et al. (2024) to decompose image representations and preserve them for further investigation. In the proposed method, the value of $\epsilon$ is set to 0.0025, 0.002, and 0.001 on ImageNet-1K, Caltech-101, and Oxford Pets, respectively. The value of $\zeta$ is set to 5. The proportion of clean validation data is set to 0.2. We use ViT-B/32 as the backbone.

### 5.2. Experimental Results

The experimental results of repairing representations and detecting backdoor samples are shown in Table 1 on ImageNet-1K, Table 4 on Caltech-101, and Oxford Pets. From these tables, we can conclude the following points.

*Table 3.* Comparison of different strategies of ablating fixed, random AHs, and reverse-ablation (denoted by "Decomp-Reverse"). "Base-Decomp" means using the original decomposed representation. "BadNet-C" ("BadNet-O") means BadNet on Caltech-101 (Oxford pets).

| Methods | BadNet | | Label Consistent | | BadCLIP | | BadNet-C | | BadNet-O | |
|---|---|---|---|---|---|---|---|---|---|---|
| | ASR | CACC | ASR | CACC | ASR | CACC | ASR | CACC | ASR | CACC |
| No Defense | 86.09 | 56.72 | 99.32 | 56.68 | 99.78 | 60.73 | 86.04 | 92.61 | 91.80 | 77.46 |
| + Base-Decomp | 88.58 | 53.71 | 87.67 | 52.87 | 99.59 | 56.28 | 90.45 | 90.51 | 94.78 | 76.80 |
| + Decomp-Rep | 21.45 | 52.25 | 17.50 | 51.42 | 0.94 | 56.08 | 4.69 | 87.95 | 34.84 | 75.00 |
| **+ Fixed [1, 2, 3]** | 86.53 | 49.72 | 87.71 | 49.42 | 99.18 | 51.78 | 82.70 | 88.93 | 94.38 | 77.18 |
| **+ Fixed [7, 8, 9]** | 88.68 | 47.86 | 88.74 | 47.51 | 58.12 | 50.18 | 86.84 | 86.07 | 92.06 | 76.12 |
| **+ Fixed [10, 11, 12]** | 88.82 | 46.72 | 88.29 | 46.72 | 99.57 | 49.78 | 90.97 | 89.64 | 96.29 | 40.91 |
| **+ Random AHs** | 72.82 | 48.30 | 77.73 | 46.16 | 82.34 | 48.86 | 70.17 | 87.34 | 83.25 | 68.31 |
| Original Clean | - | 56.72 | - | 56.68 | - | 60.73 | - | 92.61 | - | 77.46 |
| **+ Decomp-Reverse** | 47.15 | 27.85 | 39.72 | 32.42 | 80.54 | 10.07 | 32.19 | 60.51 | 18.46 | 70.23 |

*Table 4.* ASR (↓%) and CACC (↑%) comparison on Caltech-101 and Oxford Pets. "Base-Decomp" indicates using the original decomposed representation.

| Methods | Caltech-101 (accordion) | | Oxford Pets (samoyed) | |
|---|---|---|---|---|
| | ASR | CACC | ASR | CACC |
| No Defense | 86.04 | 92.61 | 91.80 | 77.46 |
| + Base-Decomp | 90.45 | 90.51 | 94.78 | 76.80 |
| **+ Decomp-Rep** | 4.69 | 87.95 | 34.84 | 75.00 |
| CleanCLIP | 31.48 | 89.55 | 70.65 | 73.73 |
| + Base-Decomp | 40.76 | 87.14 | 73.05 | 66.21 |
| **+ Decomp-Rep** | 15.51 | 86.98 | 32.76 | 66.51 |

***Basic representation decomposing has little defense effect.*** We can see that using the original representation decomposition can not significantly decrease the ASR of backdoor attacks, and even increase them in some cases (e.g., Bad-Net on ImageNet-1K). This observation implies backdoor attacks have little indirect effect on model components since representation decomposing only considers the direct effects of model components and neglects all indirect effects. Meanwhile, using representation decomposing decreases CACC slightly (i.e., CACC drops by 2%∼3%), which implies that the indirect effects of decomposing have little effect on generalization.

***Decomp-Rep achieves strong defense performance.*** Based on the basic representation decomposing, Decomp-Rep further repairs representations of heavily infected attention heads (AHs), which greatly decreases the ASR of backdoor attacks and maintains the CACC. Specifically, Decomp-Rep reduces the ASR of BadCLIP, a state-of-the-art backdoor attack, to near zero while maintaining the CACC, which verifies the superiority of Decomp-Rep. Besides, we also show the performance of repairing representations on Caltech-101 and Oxford Pets as shown in Table 4. We can see that our method also achieves superior performance. This observation implies that our method is scalable to other datasets.

***Decomp-Rep can further improve the defense performance of CleanCLIP.*** When using the fine-tuned CLIP by Clean-

CLIP, Decomp-Rep can further reduce the ASR of backdoor attacks. This observation validates the scalability of Decomp-Rep to existing defense methods (Decomp-Rep is plug-and-play with these defense methods).

***Decomp-Det achieves superior detection performance.*** We can see that Decomp-Det achieves superior performance in all cases by a significant margin. Specifically, the average AUROC performance of our method exceeds STRIP, SCALE-UP, and TeCo by 0.118, 0.220, and 0.187, respectively, which validates the superiority of Decomp-Det. Specifically, we found that Decomp-Det can achieve better detection performance against powerful backdoor attacks, e.g., BadCLIP.

### 5.3. Further analysis on repairing representations

In this section, we further analyze the proposed representation repairing method by exploring the effects of ablating different AHs and poisoning clean representations of the affected representations of selected AHs.

***Repairing representations of fixed and random AHs.*** To further validate the effectiveness of selected AHs in Decomp-Rep, we also conduct experiments of mean-ablating different fixed AHs,i.e., [1, 2, 3], [7, 8, 9], and [10, 11, 12] indicating AHs in the corresponding location of the last model layer. The experimental results are shown in Table 3. From the table, we can see that these strategies of ablating fixed AHs have a limited ability to reduce the ASR in almost all cases compared with the cases of no defense and basic representation decomposition. This observation reveals that the distribution of infected AHs is quite different in backdoor images, so we can not simply specify fixed infected AHs for all backdoor images. This is also why we use the strategy in Decomp-Rep, which detects heavily infected AHs for each image. On the other hand, ablating more random AHs achieves a slightly better performance in ASR compared with the fixed strategies, but still fails to reduce the ASR effectively.

*Table 5.* Ablation study on ImageNet-1K "w/o All AHs" means ablating all attention heads; "w/o All MLPs" means ablating all MLPs; "w Abandon" means directly replacing representations with zero values; "w Random Prototypes" means replacing representations with random values.

| Ablation | BadNet | | Blended | | Label Consistent | | ISSBA | | BadCLIP | |
|---|---|---|---|---|---|---|---|---|---|---|
| | ASR | CACC | ASR | CACC | ASR | CACC | ASR | CACC | ASR | CACC |
| w/o All AHs | 1.21 | 2.10 | 99.91 | 2.26 | 3.01 | 1.91 | 97.55 | 2.11 | 0.01 | 2.45 |
| w/o All MLPs | 88.87 | 44.83 | 0.41 | 44.56 | 88.98 | 44.35 | 1.94 | 45.05 | 99.56 | 46.05 |
| w Abandon | 44.42 | 51.66 | 0.48 | 43.28 | 34.57 | 50.64 | 2.58 | 43.46 | 63.19 | 53.12 |
| w Random Prototypes | 0.39 | 12.87 | 0.01 | 0.18 | 0.02 | 6.94 | 0.01 | 0.10 | 1.31 | 35.18 |
| **Decomp-Rep** | 21.45 | 52.25 | 0.77 | 45.25 | 17.50 | 51.42 | 6.33 | 45.68 | 25.08 | 53.72 |

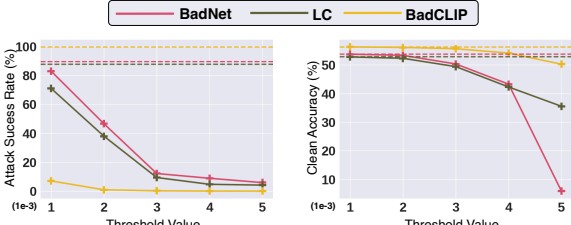

*Figure 6.* Parameter analysis on the value of $\epsilon$.

***Reversely poisoning representations of the selected AHs into clean images.*** Besides, to further validate the effect of infected AHs, we design a reverse-engineering experiment denoted by "Decomp-Reverse" that uses the representations of selected affected AHs to replace the clean representations of the same AHs in clean images. The results are shown in Table 3. From the table, we can see that using the affected representations of the selected AHs significantly increases the ASR against various backdoor attacks while reducing the CACC. This observation indicates that the selected AHs indeed contain the representation information of triggers and will construct a connection between clean images and triggers, thereby greatly increasing the ASR.

### 5.4. Parameter analysis

Here, we evaluate the value of $\epsilon$ in Eq. (5). The results are shown in Figure 6. We can see that as the value of $\epsilon$ increases, the ASR of backdoor attacks decrease gradually. This is because more attention heads will be ablated as the value of $\epsilon$ increases. However, the CACC of backdoor attacks also has a large decrease because ablating more falsely selected AHs degrades the generalization of image representations. This observation indicates that we should select affected AHs for repairing as much as possible. Therefore, it is very crucial to select the appropriate value of $\epsilon$.

### 5.5. Ablation Study

Here, we conduct the ablation study to investigate the significance of each part in our method. The results are shown in Table 5. "w/o All AHs" means ablating all attention heads. This ablation makes the ASR of BadNet, Label Consistent,

and BadCLIP reach near zero but has little effect on the ASR of Blended and ISSBA, meanwhile greatly decreasing the CACC for all backdoor attacks. On the other hand, "w/o All MLPs" means ablating all MLPs, which makes the ASR of Blended and ISSBA reach near zero but has little effect on the ASR of BadNet, Label Consistent, and BadCLIP, meanwhile slightly decreasing the CACC for all backdoor attacks. These two cases validate the necessity of selectively mean-ablating AHs and MLPs. Moreover, we will conduct an ablation study on the strategy of repairing representations of infected AHs and MLPs. Specifically, "w Abandon" means directly replacing representations with zero values. This strategy has a positive effect on decreasing the ASR compared with the basic representation decomposing (meanwhile slightly decreasing the CACC), but is still degraded compared with our strategy of using head-specific prototypes. "w Random Prototypes" means replacing representations with random values followed by a standard normal distribution. This strategy greatly decreases both the ASR and CACC of all backdoor attacks, indicating that these random values destroy the representation information. Meanwhile, this observation also indicates that it is significant to use higher-quality representations to repair representations of backdoor images. Overall, our selective ablation of AHs is a significant strategy in Decomp-Rep, which can effectively eliminate infected AHs and have little effect on other AHs.

## 6. Conclusion

In this paper, we present a comprehensive empirical study of how backdoor attacks affect CLIP. Our empirical findings reveal the attack preference of backdoor attacks on model components, the difference in the locations of infected components, and the different effects of backdoor attacks on the functionality of infected components. Inspired by these findings, we propose to repair representations of infected components or filter backdoor samples. Experimental results validate the empirical findings and the effectiveness of our methods. We hope that our findings can motivate more researchers to design effective defense methods against backdoor attacks on CLIP.

## Acknowledgment

This research is supported by the National Research Foundation Singapore and DSO National Laboratories under the AI Singapore Programme (AISGAward No: AISG2-GC-2023-009 and AISG4-GC-2023-009-1B). Feng Liu is supported by the Australian Research Council (ARC) with grant number DE240101089, LP240100101, DP230101540 and the NSF&CSIRO Responsible AI program with grant number 2303037.

## Impact Statement

Our research contributes to AI security by investigating how backdoor attacks affect CLIP, which has a positive social impact. However, we acknowledge the possibility that tricky attackers could use our findings to design specialized methods to attack CLIP. Future work should explore the robustness of our method against adaptive attacks.

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

## A. Related Works

In this section, we briefly review backdoor attacks and defenses on supervised learning and CLIP, and interpret works on CLIP's image representations.

**Backdoor attacks and defenses on supervised learning.** Backdoor attacks are serious security threats to machine learning systems (Li et al., 2022; Carlini & Terzis, 2022; Xu et al., 2022; Chen et al., 2021; Tao et al., 2024). Early research on backdoor attacks focused on designing a variety of triggers that satisfy the practical application scenarios, mainly including invisible stealthy triggers (Chen et al., 2017; Turner et al., 2019; Li et al., 2021; Doan et al., 2021; Nguyen & Tran, 2021; Gao et al., 2023; Souri et al., 2022) and physical triggers (Chen et al., 2017; Wenger et al., 2021). To defend against these attacks, researchers proposed a series of defense methods at different stages of developing models, i.e., data cleaning in the pre-processing stage (Tran et al., 2018; Zeng et al., 2023; Liu et al., 2023a; Qi et al., 2023), robust anti-backdoor training (Chen et al., 2022; Zhang et al., 2022; Huang et al., 2023), mitigation in the post-training stage (Min et al., 2023; Wang et al., 2024; Zhu et al., 2024b; Min et al., 2024; Wang et al., 2023; Xiang et al., 2022), and test-time detection in the inference stage (Shi et al., 2023; Mo et al., 2024; Guo et al., 2023; Liu et al., 2023b; Feng et al., 2023). Recently, researchers have paid much attention to the backdoor security of vision transformers and proposed customized backdoor attack and defense methods based on the characteristics of vision transformers (Yuan et al., 2023; Doan et al., 2023; Subramanya et al., 2024; Zheng et al., 2023).

**Backdoor attacks and defenses on CLIP.** As multimodal models achieve significant development, researchers have paid much attention to the backdoor security on multimodal models (Walmer et al., 2022; Han et al., 2024; Liang et al., 2024; Zhu et al., 2024a; Yang et al., 2023c; Zhu et al., 2024c; Xun et al., 2024; Huang et al., 2025; Bai et al., 2024; Hu et al., 2025; Singh et al., 2024; Yang et al., 2023a). Pioneer (Carlini & Terzis, 2022) disclosed that multimodal contrastive learning is susceptible to backdoor attacks. Furthermore, BadCLIP (Liang et al., 2023) designed a dual-embedding framework for backdoor attacks on CLIP by making visual trigger patterns approximate the textual target semantics in the embedding space. To defend against backdoor attacks, RoCLIP (Yang et al., 2023b) proposed robust multimodal contrastive learning during the pertaining stage by modifying images' captions. CleanCLIP (Bansal et al., 2023) aimed to fine-tune the backdoored CLIP by using an additional unimodal self-supervised loss. TIJO (Sur et al., 2023) focused on trigger inversion to reverse-engineer the triggers in both modalities. TA-cleaner (Xun et al., 2024) proposed to select a few samples for positive and negative subtext generation at each epoch, and align the subtexts to the images to strengthen the text self-supervision.

**Interpreting CLIP's image representations.** Although CLIP's powerful visual representation ability has achieved impressive performance on many downstream tasks, there is still a limited understanding of what information is encoded in CLIP's representations. To better understand CLIP, there were a few works that attempt to interpret visual contents by text representations, such as providing text descriptions for image regions in which a neuron is active (Hernandez et al., 2022), projecting model features into a bank of text-based concepts (Yuksekgonul et al., 2023), and studying entanglement in CLIP between images of words and natural images (Materzyńska et al., 2022). Specifically, recent work (Gandelsman et al., 2024) had a further exploration of CLIP's image representations by decomposing them into text-explainable directions that are attributed to specific attention heads and image locations. Similarly, INViTE (Chen et al., 2024) presented a framework for interpreting ViT's latent tokens with text explanations.

## B. Details of datasets

### B.1. Evaluation Datasets

In this paper, we evaluate attack success rates and clean accuracy on three downstream datasets: ImageNet-1K (Russakovsky et al., 2015), Caltech-101 (Fei-Fei et al., 2004), and Oxford Pets (Parkhi et al., 2012). The target classes on ImageNet-1K, Caltech-101, and Oxford Pets are "banana", "accordion", and "Samoyed" respectively. Besides, we select clean image-text pairs from CC3M (Sharma et al., 2018) to fine-tune the backdoored CLIP. Here, we will introduce the details of these datasets.

- ImageNet-1K consists of 1,000 classes and over a million images, making it a challenging dataset for large-scale image classification tasks.

- Caltech-101 contains 101 object categories and 1 background category with 40 to 800 images per category, which are both commonly used for testing model performance on fine-grained classification and image recognition tasks.

- Oxford Pets is a 37-category pet dataset with roughly 200 images for each class created by the Visual Geometry Group

| | | | |
|---|---|---|---|
| A droplet in motion | A ball A bamboo | A low-resolution image | An image of a Engineer |
| advanced artificial intelligence | Abandoned factory space | A magnet | An image of a entree |
| advanced biotechnology | Abandoned spaces | A magnolia | An image of a face |
| advanced drone technology | A barbed wire design | A marbled texture | An image of a family |
| advanced renewable energy | A barcode | A marsh | An image of a Farmer |
| advanced robotics | A basket | A mask | An image of a Fashion Designer |
| advanced robotic technology | A beam | A maze | An image of a Film Director |
| advanced space exploration | A beautiful photo | A meadow | An image of a Financial Analyst |
| advanced transportation | A belt | A meandering river | An image of a Firefighter |
| advanced transport system | A bicycle | A megaphone | An image of a Flight Attendant |
| Adventurous explorations | A blade | A meteor | An image of a Florist |
| Advertisment | A blade (of a fan or a saw) | A microphone | An image of a Gardener |
| A earring | A blade (of grass or a knife) | A mirror | An image of a Graphic Designer |
| Aerial landscape photography | A blanket | A modular structure | An image of a Gymnast |
| Aerial perspective | A blurry image | Ancient and weathered artifact | An image of a Hair Stylist |
| Aerial view | A bolt | Ancient and weathered stone carving | An image of a head |
| Aerial view of a bay | A bonnet | Ancient and weathered stone structure | An image of a IT Specialist |
| Aerial view of a bustling metropolis | A book | Ancient castle walls | An image of a Journalist |
| Aerial view of a cityscape | A bookmark | Ancient historical site | An image of a Judge |
| Aerial view of a coastal area | A boot | Ancient ruins | An image of a king |
| Aerial view of a construction site | A bottle | Ancient temple ruins | An image of a lake |
| Aerial view of a coral reef | A bowl | An equilateral hexagon | An image of a Landscaper |
| Aerial view of a countryside | A bracelet | An equilateral pentagon | An image of a Lawyer |
| Aerial view of a desert oasis | A branch | An equilateral triangle | An image of a Librarian |
| Aerial view of a farmland | A breeze | Angry facial expression | An image of a main course |
| Aerial view of a hamlet | A brick | An illustration of an animal | An image of a Marine Biologist |
| Aerial view of a harbor | A brush | An image capturing an interaction | An image of a Mechanic |
| Aerial view of a inlet | Abstract acrylic painting | between subjects | An image of a Musician |
| Aerial view of a marketplace | Abstract artwork with concentric circles | An image of a Accountant | An image of a Music Producer |
| Aerial view of a mountain range | Abstract artwork with cross-hatching | An image of a Aerospace Engineer | An image of Andorra |
| Aerial view of an agricultural field | Abstract artwork with splatter paint | An image of a Animal Trainer | An image of a Novelist |
| Aerial view of an archaeological site | Abstract artwork with swirls | An image of a Arborist | An image of a Nurse |
| Aerial view of a natural landscape | Abstract composition | An image of a Archaeologist | An image of a Swimmer |
| Aerial view of an industrial area | Abstract expressionist artwork | An image of a Architect | An image of a Systems Analyst |
| Aerial view of an island | Abstract form | An image of a Art Historian | An image of a Teacher |
| Aerial view of an ocean coastline | Abstract geometric patterns | An image of a Artist | An image of a Veterinarian |
| Aerial view of an urban skyline | abstract geometric shapes | An image of a Astronomer | An image of a Waiter/Waitress |
| Aerial view of a paradise | abstract graffiti | An image of a Athlete | An image of a Welder |
| Aerial view of a promenade | Abstract oil painting | An image of a Attorney | An image of a Writer |
| Aerial view of a river or stream | Abstract patterns | An image of a Auto Mechanic | An image of a Zoologis |
| Aerial view of a serene countryside | Abstract reflections | An image of a Ballet Dancer | Tranquil atmospheres |
| | | | Time-worn beauty |

*Figure 7.* Examples of used text descriptions.

at Oxford. The images have large variations in scale, pose, and lighting. All images have an associated ground truth annotation of breed, head ROI, and pixel-level trimap segmentation.

• CC3M[1] is a dataset consisting of about 3.3M images annotated with captions. In contrast with the curated style of other image caption annotations, Conceptual Caption images and their raw descriptions are harvested from the web, and therefore represent a wider variety of styles. More precisely, the raw descriptions are harvested from the Alt-text HTML attribute associated with web images.

### B.2. Text descriptions

To characterize the functionality of model components, we employed TEXSPAN proposed by (Gandelsman et al., 2024). The algorithm needs a pool of candidate text descriptions. Specifically, they prompted ChatGPT (GPT-3.5) to produce image descriptions. The prompt was "Imagine you are trying to explain a photograph by providing a complete set of image characteristics. Provide generic image characteristics. Be as general as possible and give short descriptions presenting one characteristic at a time that can describe almost all the possible images of a wide range of categories. Try to cover as many categories as possible, and don't repeat yourself. Here are some possible phrases: "An image capturing an interaction between subjects", "Wildlife in their natural habitat", "A photo with a texture of mammals", "An image with cold green tones", "Warm indoor scene", "A photo that presents anger". Just give the short titles, don't explain why, and don't combine two different concepts (with "or" or "and"). Make each item in the list short but descriptive. Don't be too specific." This process resulted in 3498 sentences as shown in Figure 7.

---

[1] https://huggingface.co/datasets/pixparse/cc3m-wds

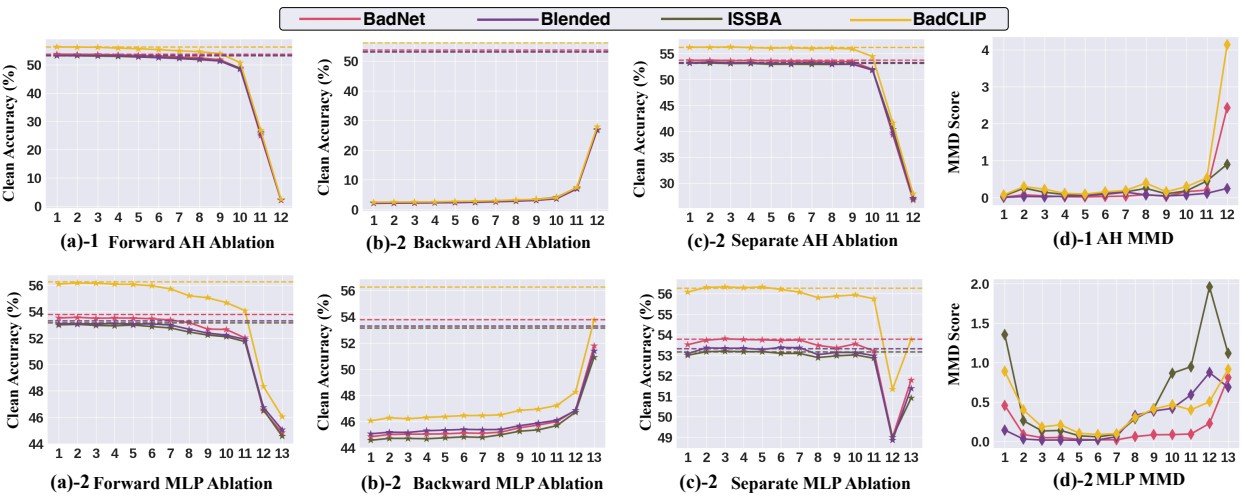

*Figure 8.* Mean-ablation on model components. Figures (a)-1/2, (b)-1/2, and (c)-1/2 show the CACC of forward, backward, and separate ablation on AHs/MLPs, respectively. Figures (d)-1/2 show the layer-wise MMD on AHs and MLPs, respectively. Dashed lines indicate the baseline CACC of backdoor attacks. Best viewed in color.

## C. Pseudo-code of our proposed method

---
**Algorithm 1** Our methods of repairing representations or filtering backdoor samples

---
**Input:** a backdoored CLIP $\{\widetilde{\mathcal{V}}(\cdot), \widetilde{\mathcal{T}}(\cdot)\}$, similarity threshold $\epsilon$, detection threshold $\zeta$, test data $\mathcal{X}_{test}$, validation data $\mathcal{X}_{val}$;
1: Construct head-specific prototypes $\Phi_h$ on the validation data $\mathcal{X}_{val}$;
2: Construct MLP-specific prototypes $\Phi_m$ on the validation data $\mathcal{X}_{val}$;
3: **for** $x_i$ in $\mathcal{X}_{test}$ **do**
4:     **if** Blended or ISSBA **then**
5:         Replace the representations of the last five MLPs with MLP-specific prototypes;
6:     **else**
7:         Use the detector $\Psi$ in Eq. (5) to find infected attention heads;
8:         Count the number of infected attention heads and use the detector $\omega$;
9:         Replace the representations of selected AHs with those of head-specific prototypes;
10:     **end if**
11: **end for**
12: Calculate ASR, CACC, or AUROC;
13: Output the metrics.

---

## D. Detailed settings

### D.1. Detailed settings of backdoor attacks

In the experiment, we use five backdoor attacks: BadNet (Gu et al., 2017), Blended (Chen et al., 2017), Label Consistent (Turner et al., 2019), ISSBA (Li et al., 2021), and BadCLIP (Liang et al., 2023). Here, we introduce these methods in detail.

- BadNet[2] is a seminal work on backdoor attacks in deep learning, generating poisoned examples by stamping a small patch randomly into images and altering their labels to the target class. We set the patch size to 16 pixels.

- Blended enhances the stealthiness of backdoor attacks from the perspective of the trigger. It implements an invisible backdoor attack by blending the trigger with the original images linearly, thus evading human detection. The blending ratio for the trigger is 0.2.

- Label Consistent enhances the stealthiness of backdoor attack from the perspective of the label. It employs generative

---
[2]https://github.com/THUYimingLi/BackdoorBox

models or adversarial perturbations to selectively poison images associated with the target class.

- ISSBA[3] introduces an invisible attack that creates sample-specific triggers by encoding an attacker-specified string into benign images using an encoder-decoder network.

- BadCLIP[4] proposes a backdoor attack on CLIP, which optimizes visual trigger patterns in a dual-embedding guided framework to make the attack undetectable. For BadCLIP, we employ the same parameter settings specified in the original paper.

For these backdoor attacks, we utilize the AdamW optimizer with an initial learning rate of 1e-5, applying cosine scheduling over a total of five epochs with a batch size of 128.

### D.2. Detailed settings of CleanCLIP

CleanCLIP[5] (Bansal et al., 2023) defends against backdoor attacks in multimodal contrastive learning by optimizing the integration of multimodal contrastive and unimodal self-supervised losses using a limited amount of clean data. Note that the backbone of the visual encoder in CleanCLIP is ResNet-50. In this paper, we use the vision transformer (ViT-B/32) as the visual encoder. We adapted the parameters used in the original paper to our case. Specifically, we randomly selected 10,0000 image-text pairs from CC3M as the fine-tuning data. The learning rates were set to 5e-6 for BadNet, Blended, and BadCLIP, and 3e-6 for Blended and ISSBA on ImageNet-1K. The batch size was 64. The fine-tuning epoch was 10. Note that we did not blindly reduce attack success rates by adjusting the learning rates, but maintained clean accuracy of the fine-tuned model.

### D.3. Detailed settings of detection methods

In the experiment, we compare three backdoor detection methods: STRIP (Gao et al., 2019), SCALE-UP (Guo et al., 2023), and TeCo (Liu et al., 2023b). Here, we introduce these methods in detail.

- STRIP[6] is the first black-box TTSD method that overlays various image patterns and observes the randomness of the predicted classes of the perturbed input to identify poisoned samples. In our experiments, for each input image, we use 64 clean images from the test data for superimposition.

- SCALE-UP[7] is also a method for black-box input-level backdoor detection that assesses the maliciousness of inputs by measuring the scaled prediction consistency (SPC) of labels under amplified conditions, offering effective defense in scenarios with limited data or no prior information about the attack.

- TeCo[8] modifies input images with common corruptions and assesses their robustness through hard-label outputs, ultimately determining the presence of backdoor triggers based on a deviation measurement of the results. In our experiments, considering concerns about runtime, we selected "elastic_transform", "gaussian_noise", "shot_noise", "impulse_noise", "motion_blur", "snow", "frost", "fog", "brightness", "contrast", "pixelate", and "jpeg_compression" as methods for corrupting images. The maximum corruption severity was set to 6.

## E. Details of TEXTSPAN

The objective of TEXTSPAN (Gandelsman et al., 2024) is to find descriptive texts of a candidate text pool for the model component. To this end, TEXTSPAN employs a greedy algorithm to identify a set of $m$ descriptions for each head that can span its output space [9].

- (1) It first constructs a matrix $C^{(l,h)}$ denoted by the head outputs for head $(l, h)$, and a matrix $\mathcal{T}$, which contains the representations of the candidate descriptions $\{t_i\}_{i=1}^{M}$ projected onto the span of $C$.

---

[3]https://github.com/yuezunli/ISSBA
[4]https://github.com/LiangSiyuan21/BadCLIP
[5]https://github.com/nishadsinghi/CleanCLIP
[6]https://github.com/garrisongys/STRIP
[7]https://github.com/JunfengGo/SCALE-UP
[8]https://github.com/CGCL-codes/TeCo
[9]https://github.com/yossigandelsman/clip_text_span

- (2) In each iteration, the algorithm calculates the dot product between each row of $\mathcal{T}$ and the head outputs $C$, identifying the row with the highest variance, $\mathcal{T}[j^*]$ (the first "principal component").

- (3) It then removes the contribution of this component from all rows and repeats the process to discover the next components. This projection ensures that each new component contributes variance orthogonal to the previous ones.

## F. Limitation

We present two limitations of our investigation. First, the representation decomposing ignores the indirect effects of model components on the representation, e.g., information flow from early layers to deeper ones, which may loss some generalization information compared with the original representation. Second, we focus on qualitatively characterizing the change in the functionality of attention heads caused by backdoor attacks, which may require certain quantitative metrics.

