# OpenReview forum: "A Closer Look at Backdoor Attacks on CLIP"
_ICML.cc/2025/Conference — ICML 2025 poster_

### Official Review · Reviewer_gzt8 · 2025-03-08

**Overall Recommendation:** 3

**Summary:**

The paper presents a detailed analysis of which type/location of layers are effected by backdoor attacks in transformer based VLMs by the help of representation decomposing. The findings indicate: global trigger based attacks mostly affect MLPs whereas localized trigger patches influence Attention heads. Based on this analysis a detection method termed Decomp-Rep which uses representation decomposing. The method seems to improve backdoor cleaning rate

**Claims And Evidence:**

For the architecture (ViT-B/32) considered in the work: the evidence seems sufficient for this.

**Essential References Not Discussed:**

RoCLIP[1] also uses candidate texts from a retrieval pool to clean backdoored models, currently only mentioned in related work but not compared to.
BDetCLIP[2], a recent detection method: works on a similar setting - just added in related work not compared to: this comparison would be the most relevant
Ta-cleaner[3]: not discussed, this is similar to CleanCLIP, and according to the paper, better than CleanCLIP.


[1] Yang, W.,et al. Robust  contrastive language-image pretraining against data poisoning and backdoor attacks. In NeurIPS, 2023.
[2] Niu, Yuwei, et al. "Bdetclip: Multimodal prompting contrastive test-time backdoor detection." arXiv preprint arXiv:2405.15269 (2024).
[3] Xun, Yuan, et al. "Ta-cleaner: A fine-grained text alignment backdoor defense strategy for multimodal contrastive learning." arXiv preprint arXiv:2409.17601 (2024).

**Experimental Designs Or Analyses:**

The experiments are only considered in backdoor learning by fine-tuning. I am not sure if the same conclusions would hold if backdoors were injected while training from scratch.

**Methods And Evaluation Criteria:**

The backdoor learning is done in the fine-tuning setup and different attacks are considered. These seems sufficient for the problem.

**Other Comments Or Suggestions:**

Links broken on line 251, 651 etc

**Other Strengths And Weaknesses:**

Strengths:
The initial experiments to find how the backdoors effect certain layers is very nice and commendable. The way contributions are highlighted and explained also is a plus point.

Weaknesses:
- The final experiments for the 'Decomp-Rep' seem lacking. The tests are only on ViT-B/32 (what would happen for more patch models like B/16? or larger ones like L/14), the baselines are not sufficient.

- Decomp-Rep seems to clean better than the competitors, but also looses CACC, for instance on BadNet for CLeanCLIP: the ASR with
Decomp-Rep goes down to 41.49% from 54.23%, but the decay in CACC is 6%. If the detection method is unable to clean to ASR levels around 10-20% then this decay in CACC is too large. For the Base model case as well the decay in clean performance is too large (upto 10% in some cases). Ideally one wants a completely clean model with a small degradation in clean performance, but Decomp-Rep seems to not do it from the results.From fig. * it seems BadNet and LC attacked models are not useful after detection with Decomp-Rep.
- The only attack for which Decomp-Rep seems to work reasonable well is Blended. I think just improving a bit on some baselines is not enough to call Decomp-Rep effective.

**Questions For Authors:**

1. The tests are only on ViT-B/32, what would happen for more patch models like B/16? or larger ones like L/14?
2. Why no comparison to methods like BDetCLIP[1], which is also a detection method?

Overall, even a positive response to these questions might not change my evaluation of this work, as I am not convincesd by its effectiveness. The method is only slightly better than baselines and completely cleaning backdoors seems not possible for the method without completely destroying the CACC.

[1] Niu, Yuwei, et al. "Bdetclip: Multimodal prompting contrastive test-time backdoor detection." arXiv preprint arXiv:2405.15269 (2024).

**Relation To Broader Scientific Literature:**

in my opinion the initlal experiments on finding which attacks effect which layers is novel.
The proposed method is based on detection and uses text descriptions. There are other detection methods in literature but this method seems different to those.

**Theoretical Claims:**

There are no theoretical claims only notation used to motivate the representation decomposition and that looks okay.

---

> ### Author Rebuttal · Authors · 2025-04-01
>
> We sincerely thank you for your insightful comments. We are encouraged by your recognition of the novelty and the solid experimental analysis of our work. Below are your concerns and our corresponding responses:
>
> **Q1: The experiments of backdoor learning from scratch.**
>
> **A:** Thank you for your valuable comment. To solve your concern, we have conducted an additional experiment of training a backdoored CLIP in the pre-training stage. Specifically, following the attack setting in CleanCLIP, we randomly select 1,500 image-text pairs from CC3M as backdoor samples (the target class is ''banana'' and we use the triggers from BadNet and Blended). Then, we train CLIP from scratch on the backdoored CC3M dataset and analyze the backdoored CLIP. The experimental results are shown in the following tables.
>
> **Table: AH ablation (BadNet)**
> |1-4|5-8|9|10|11|12|
> |:--:|:--:|:--:|:--:|:--|:--:|
> |97.32|96.86|95.46|95.78|96.51|1.79|
>
> **Table: MLP ablation (Blended)**
> |1-4|5-8|9|10|11|12|13|
> |:--:|:--:|:--:|:--:|:--|:--:|:--:|
> |98.35|97.67|95.39|92.72|68.51|20.42|5.67|
>
> From the tables, we can see that BadNet (Blended) also mainly affects AHs (MLPs) in the pre-trained backdoored CLIP, which is consistent to our claim.
>
> **Q2: Missing TA-cleaner.**
>
> **A:** Thank you for pointing out the related paper. We will discuss it in the related work.
>
> **Q3: Comparing RoCLIP and BDetCLIP**
>
> **A:** Thank you for your valuable comment. We would like to explain that RoCLIP aims to mitigate backdoor implanting during the pre-training stage, while our proposed method focuses on eliminating the implanted backdoor in the pre-trained CLIP, which works in different stages. Despite this, we really understand your concern and have conducted additional experiments by adapting RoCLIP (ViT) to the fine-tuning stage.
>
> **Table: Comparing RoCLIP (BadNet, ASR/CACC)**
> |No Defense|+RoCLIP|+Decomp-Rep|+Both|
> |:--:|:--:|:--:|:--:|
> |86.09/56.72|75.42/54.31|21.45/52.25|48.19/51.27|
>
> From the table, we can see that our method is more effective in reducing the ASR. In addition, we also have conducted additional experiments to compare BDetCLIP (following the original setting in the paper).
>
> **Table: Comparing BDetCLIP (AUROC)**
> |Method|BadNet|BadCLIP|
> |:--:|:--:|:--:|
> |BDetCLIP|0.970|0.910|
> |Decomp-Det|0.920|0.990|
>
> From the table, we can see that although the performance of our method against BadNet is slightly lower than BDetCLIP, the performance against BadCLIP (more advanced) is superior. Moreover, our method does not require abundant class description texts from large language models, which are often time-consuming to collect.
>
> **Q4: Experiments on other model architectures.**
>
> **A:** Thanks for your valuable suggestion. We have conducted additional experiments on ViT-L/14.
>
> **Table: AH ablation (ASR)**
> |1-8|9-16|16-20|21|22|23|24|
> |:--: |:--: |:--: |:--: |:--: |:--: |:--: |
> |96.44|95.72|94.38|92.78|91.20|91.33|2.53|
>
> From the table, we can see that only ablating AHs in the last layer significantly decreases the ASR, which indicates that the infected AHs are still centered in the last layer. This observation also supports our claim.
>
> **Q5: The effectiveness of Decomp-Rep.**
>
> **A:** We would like to justify the effectiveness of Decomp-Rep. Specifically, we argue that the CACC loss mainly stems from basic representation decomposition (Base-Decomp) in Eq. (4) rather than our proposed repairing method. For instance, in Table 1, the CACC of the Base-Decomp decreased by 3.01\%, 5.26\%, and 4.45\% for BadNet, LC, and BadCLIP on the base model, respectively, and our Decomp-Rep only slightly decreases the CACC by 1.46\%, 1.45\%, and 0.2\% compared with Base-Decomp. A similar observation can also be found for the CleanCLIP model. Hence, the CACC decrease is mainly caused by the basic representation decomposition framework and can be mitigated by different representation decomposition strategies (which can be seamlessly compatible with Decomp-Rep). To show this, we have conducted additional experiments that only decompose the representations in the last layer.
>
> **Table: Comparison (ASR/CACC)**
> |No Defense|+Base-Decomp-new|+Decomp-Rep-new|
> |:--:|:--:|:--:|
> |86.09/56.72|87.42/55.31|22.08/54.82|
>
> From the table, we can see that using a different basic representation decomposition strategy can effectively alleviate the CACC decrease in our method. Notably, we would like to emphasize that *our core contributions mainly lie in pioneering a deep exploration of backdoor attacks on CLIP, revealing significant key findings, and validating them by designing lightweight defense methods instead of pursuing strictly SOTA methods*. We think that the empirical performances of Decomp-Rep and Decomp-Det have sufficiently validated our key findings, which would motivate more researchers to design more powerful defense counterparts based on these findings in the future.
>
> **Q6: Broken Links.**
>
> **A:** Thanks for pointing out the issues. We will fix them and double-check the paper.

---

> > ### Comment · Reviewer_gzt8 · 2025-04-04
> >
> > I thank the authors for the detailed replies.
> > The BDetCLIP experiment shows the proposed method is not effective under all attacks, but overall I think the work has a valid enough contribution to the community. Hence, I update my score, and lean towards a marginal accept.

---

> > > ### Author Response · Authors · 2025-04-04
> > >
> > > We sincerely appreciate your insightful recognition of our work's contributions to the community. While Decomp-Det is not strictly superior to BDetCLIP under all attacks, it remains competitive and offers valuable insights into detecting backdoor samples by analyzing specific model components. As we previously stated, our goal is not to achieve SOTA performance across all settings but rather to encourage researchers to explore more interpretable and transparent actions on specific model components to defend agianst backdoor attacks. Once again, we truly appreciate your affirmation of our valid enough contribution to the community.

---

### Official Review · Reviewer_pmEr · 2025-03-11

**Overall Recommendation:** 4

**Summary:**

This paper presents a comprehensive empirical study to analyze the effects of backdoor attacks on CLIP. They found three empirical findings about how different types of backdoor attacks have various effects on CLIP. The authors conducted extensive experiments and showed visualized results, which validates their claims.

**Claims And Evidence:**

Yes, the claims of three findings are supported by extensive empirical experiments and visualized results.

**Essential References Not Discussed:**

The related works cover backdoor attacks and defenses.

**Experimental Designs Or Analyses:**

Yes, I have checked the soundness of experimental designs and analyses. The paper compares six backdoor attacks on CLIP, which are divided into two lines: local patch-based and global perturbation-based backdoor attacks, and visualizes the attention heads of local patch-based backdoor attacks.

**Methods And Evaluation Criteria:**

Yes, the visualized results of attention heads are reasonable to characterize the backdoor effects on CLIP. Using the text descriptions to show the property role of components in CLIP is reasonable and interesting.

**Other Comments Or Suggestions:**

See the above weakness

**Other Strengths And Weaknesses:**

Strengths:
1. Originality: The paper proposes three key findings that are novel in the context of backdoor attacks on CLIP, especially for using text descriptions to interpret the backdoor effects on CLIP.
2. Clarity: The writing of the paper is good. The three key findings are clearly explained.
3. Significance: The paper has a positive contribution to the research of backdoor attacks and defenses.
4. Experimental validation: The experimental results are sufficient, including two different defense methods.

Weaknesses:
1. More experimental results should be further analyzed.
2. More different model architectures should be exploited.

**Questions For Authors:**

1. Could you further explain the results in Figure 4?

**Relation To Broader Scientific Literature:**

The key findings in the paper are related to the attention mechanism and representation learning in vision transformers. Specifically, local patch-based triggers are implanted into the attention heads in the last model layers, which are strongly related to the research of interpreting image representations in vision transformers.

**Theoretical Claims:**

N/A. The paper does not involve theoretical analysis.

---

> ### Author Rebuttal · Authors · 2025-04-01
>
> We sincerely thank you for your valuable comments. We are encouraged by your recognition of the novelty and significance of our work. Below are your concerns and our corresponding responses:
>
> **Q1: More experimental results should be further analyzed.**
>
> **A:** Thank you for your valuable comment. Following your suggestion, we will further provide a more thorough analysis on the experimental results in the ablation study and the comparison of different repairing strategies. As shown in Table 3 in the appendix, we compared three ablation experiments, including directly replacing all AHs, replacing the representations with zero values and random prototypes. From the experimental results, we can see that these three strategies are all inferior compared with our method. Specifically, directly replacing all AHs or MLPs reduces both the ASR and the CACC greatly, which is inapplicable in real-world applications. This observation validates the necessity and effectiveness of selectively repairing AHs or MLPs. Next, replacing the representations with zero values (rather than prototypes) reduces the ASR to a certain extent but still maintains a high level of ASR and a normal level of CACC. In contrast, replacing the representations with random values reduces both the ASR and the CACC greatly. This observation validates the significance of prototypes. Overall, these experiments validate the effectiveness of our method.
>
> In addition, as shown in Table 4 in the appendix, we compared repairing different combinations of AHs, including fixed and random AHs. From the experimental results, we can see that replacing fixed or random three AHs both has little effect on reducing the ASR. This observation indicates that the infected components are diverse and have no fixed preference, which reveals the challenge of repairing selection.
>
> **Q2: More different model architectures should be exploited.**
>
> **A:** Thank you for your insightful comment. We have conducted additional experiments on the CLIP with the ViT-L/14 architecture (14x14 patch size and 24 layers). The experimental results are shown in the following table.
>
> **Table: AH ablation (ASR)**
> |1-8|9-16|16-20|21|22|23|24|
> |:--:|:--:|:--:|:--:|:--:|:--:|:--:|
> |96.44|95.72|94.38|92.78|91.20|91.33|2.53|
>
> From the table, we can see that only ablating AHs in the last layer significantly decreases the ASR, which indicates that the infected AHs are still centered in the last layer. This observation also supports our claim.
>
> **Q3: Could you further explain the results in Figure 4?**
>
> **A:** Sure, we would like to explain that Figure 4 shows the descriptive texts on MLPs. Specifically, these descriptive texts are derived via the TextSpan algorithm, which indicates the semantics of MLPs in CLIP's text spaces. We select top-5 results for better visualization. In the figure, we can see that the descriptive texts on MLPs may not show certain specific property roles (e.g., color and location) like those on AHs due to their inherent architecture difference. To better quantitatively characterize the descriptive texts, we calculate the average text similarity of the descriptive texts between clean (green color) and backdoored (red color) MLPs, which reflects how backdoor attacks affect the semantics of MLPs in CLIP's text spaces. Based on the statistics, we can see that the last several MLPs have lower similarity values compared with the first ones, which further validates the observation in Figure 2 (infected MLPs are dispersed on the several late layers) from a different perspective.

---

> > ### Comment · Reviewer_pmEr · 2025-04-04
> >
> > I appreciate the authors' responses in the rebuttal. The authors' responses have effectively addressed my concerns, and I will maintain my acceptance rating.

---

> > > ### Author Response · Authors · 2025-04-05
> > >
> > > Thank you for letting us know that your concerns were addressed by our rebuttal. We sincerely appreciate that you keep the "acceptance" recommendation for our paper!

---

### Official Review · Reviewer_Yt4Z · 2025-03-15

**Overall Recommendation:** 3

**Summary:**

This paper investigates how backdoor attacks infect different components of a ViT-based CLIP model (notably attention heads vs. MLP layers) and proposes a “repair” mechanism that selectively ablates or replaces infected representations in the last few layers. The authors conduct detailed experiments to show that different attack strategies target either local patch-based features (primarily encoded in attention heads) or global perturbations (often captured by MLPs). Based on these observations, they design an approach to detect and repair “infected” attention heads or MLPs during inference.

**Claims And Evidence:**

Yes.

**Essential References Not Discussed:**

No.

**Experimental Designs Or Analyses:**

Yes.

**Methods And Evaluation Criteria:**

Yes.

**Other Comments Or Suggestions:**

No.

**Other Strengths And Weaknesses:**

**Strengths**

1. Detailed Empirical Analysis: Overall I like the paper. The paper provides a thorough layer-by-layer decomposition of ViT representations, highlighting how certain attacks affect different parts of the model.

2. Comprehensive Experiments: The authors evaluate multiple backdoor attacks (e.g., BadNet, Blended, ISSBA) and show quantitative improvements in attack success rate (ASR) reduction.

3. Novelty in Layer-wise “Repair”: Proposing a targeted approach—rather than blindly dropping entire layers—could, in principle, preserve clean accuracy.

**Weaknesses**

1. Necessity of “Partial Repair” vs. Simple Replacement: The paper emphasizes a “repair” mechanism for only a subset of attention heads. However, it is unclear why one cannot simply replace all attention heads (or MLP modules) in the final layers with known clean versions. If we have access to a small clean set and the original architecture, fully swapping suspect components might be more straightforward and yield stronger guarantees than partial repair. The paper does not sufficiently justify why partial repair is needed or superior.

2. Adaptive Attack Analysis: The proposed method might be vulnerable if attackers specifically design triggers to blend across multiple heads and layers. The paper does not show robust experiments on truly adaptive adversaries who might anticipate “repair” or partial ablation.

3. Comparisons with Simpler Baselines: While the authors compare with other detection strategies, they do not fully demonstrate that partial repair is unequivocally better than simpler alternatives (e.g., just removing final-layer attention heads altogether, or fine-tuning them from scratch, or replace all heads with clean versions).

4. Missing related work: The paper does not discuss the recent backdoor attack[1] on CLIP. It would be better to include discussion if the current observation still holds on it.

[1] Distribution preserving backdoor attack in self-supervised learning. Tao et al. IEEE S&P 2024.

**Questions For Authors:**

Please respond to each point of Weaknesses.

**Relation To Broader Scientific Literature:**

The paper is related to the safety and security of multi-modal foundation models.

**Theoretical Claims:**

No theoretical proof.

---

> ### Author Rebuttal · Authors · 2025-04-01
>
> We sincerely thank you for your valuable comments. We appreciate your recognition of the novelty of our work and the empirical analysis in the paper. Below are your concerns and our corresponding responses:
>
> **Q1: Necessity of “Partial Repair” vs. Simple Replacement**
>
> **A:** Thanks for your insightful comment. Actually, we have conducted this experiment in the ablation study. Specifically, we directly replace all AHs as a simple replacement. The experimental results are shown in Table 3 (in the Appendix due to the space limit).
>
> **Table: Compassion (ASR/CACC)**
> |Attacks|BadNet|LC|BadCLIP|
> |:--:|:--:|:--:|:--:|
> |Repair all AHs|1.21/2.10|3.01/1.91|0.01/2.45|
> |Ours|21.45/52.25|17.50/51.42|25.08/53.72|
>
> From the table, we can see that repairing all AHs reduces both the ASR and CACC nearly to zero. This observation indicates that although we can directly remedy all suspect components to nearly totally eliminate the backdoor, the caused side effect is unaffordable, i.e., the CACC is too low. Therefore, we have to selectively repair certain AHs to achieve a better tradeoff between ASR and CACC, which is exactly what our proposed method does.
>
> **Q2: Adaptive Attack Analysis.**
>
> **A:** Thanks for your insightful comment. We would like to explain that the adversary generally *cannot* control the trigger to specifically blend across multiple heads and layers under the black-box setting. We really understand your concern about the robust experiments against this specific adaptive attack. Hence, we have conducted additional experiments for this purpose. Specifically, we suppose that the adversary can implant the trigger into specific AHs (i.e., 1-th, 3-th, and 5-th) in certain layers (i.e., the last layer and the second-to-last layer) by replacing the clean representations of these components with the infected ones. In this way, we can use our proposed method to defend against this adaptive attack. The experimental results are shown in the following table.
>
> **Table: Adaptive attack and defense (BadNet)**
> |Method|ASR|CACC|
> |:--:|:--:|:--:|
> |Attack|82.43|54.78|
> |Defense|48.16|52.35|
>
> From the table, we can see that our proposed method can effectively defend against the adaptive attack. This observation indicates our method is robust to specifically designed backdoor attacks on specific AHs.
>
> **Q3: Comparisons with Simpler Baselines**
>
> **A:** Thanks for your valuable comment. Actually, we have conducted experiments with different AH repairing strategies. Specifically, we use the fixed AH strategy on three types of AHs and the random strategy. The experimental results are from Table 4 in the appendix.
>
> **Table: Different strategies (ASR/CACC)**
> |Strategy|BadNet|LC|BadCLIP|
> |:--:|:--:|:--:|:--:|
> |Fixed[1,2,3]|86.53/49.72|87.71/49.42|99.18/51.78|
> |Fixed[7,8,9]|88.68/47.86|88.74/47.51|58.12/50.18|
> |Fixed[10,11,12]|88.82/46.72|88.29/46.72|99.57/49.78|
> |Random|72.82/48.30|77.73/46.16|82.34/48.86|
> |Ours|21.45/52.25|17.50/51.42|0.94/56.08|
>
> From the table, we can see that the fixed and random repairing strategies cannot effectively reduce the ASRs and lose more CACCs. This is because they cannot find infected components effectively. In contrast, our method can specifically target infected components and repair their representations effectively.
>
> **Q4: Missing related work.**
>
> **A:** Thank you for pointing out the related paper. We will discuss the paper in the related work.

---

> > ### Comment · Reviewer_Yt4Z · 2025-04-04
> >
> > Thanks for the rebuttal. I kept my score as "weak accept".

---

> > > ### Author Response · Authors · 2025-04-05
> > >
> > > We sincerely thank you for your positive assessment of our paper. Your valuable suggestions will significantly help us improve the quality and clarity of our work.

---

### Decision · Program_Chairs · 2025-05-01

**Decision:**

Accept (poster)

**Comment:**

The rebuttal was convincing for most reviewers, so all reviewers suggested accepting it. Hence, it will be accepted.